# MONOTONIC CHUNKWISE ATTENTION

**Chung-Cheng Chiu** [*] **& Colin Raffel** [*]
Google Brain
Mountain View, CA, 94043, USA
`{chungchengc,craffel}@google.com`

## ABSTRACT

Sequence-to-sequence models with soft attention have been successfully applied to a wide variety of problems, but their decoding process incurs a quadratic time and space cost and is inapplicable to real-time sequence transduction. To address these issues, we propose Monotonic Chunkwise Attention (MoChA), which adaptively splits the input sequence into small chunks over which soft attention is computed. We show that models utilizing MoChA can be trained efficiently with standard backpropagation while allowing online and linear-time decoding at test time. When applied to online speech recognition, we obtain state-of-the-art results and match the performance of a model using an offline soft attention mechanism. In document summarization experiments where we do not expect monotonic alignments, we show significantly improved performance compared to a baseline monotonic attention-based model.

## 1 INTRODUCTION

Sequence-to-sequence models (Sutskever et al., 2014; Cho et al., 2014) with a soft attention mechanism (Bahdanau et al., 2015) have been successfully applied to a plethora of sequence transduction problems (Luong et al., 2015; Xu et al., 2015; Chorowski et al., 2015; Wang et al., 2017; See et al., 2017). In their most familiar form, these models process an input sequence with an encoder recurrent neural network (RNN) to produce a sequence of hidden states, referred to as a memory. A decoder RNN then autoregressively produces the output sequence. At each output timestep, the decoder is directly conditioned by an attention mechanism, which allows the decoder to refer back to entries in the encoder's hidden state sequence. This use of the encoder's hidden states as a memory gives the model the ability to bridge long input-output time lags (Raffel & Ellis, 2015), which provides a distinct advantage over sequence-to-sequence models lacking an attention mechanism (Bahdanau et al., 2015). Furthermore, visualizing where in the input the model was attending to at each output timestep produces an input-output alignment which provides valuable insight into the model's behavior.

As originally defined, soft attention inspects every entry of the memory at each output timestep, effectively allowing the model to condition on any arbitrary input sequence entry. This flexibility comes at a distinct cost, namely that decoding with a soft attention mechanism has a quadratic time and space cost $\mathcal{O}(TU)$, where $T$ and $U$ are the input and output sequence lengths respectively. This precludes its use on very long sequences, e.g. summarizing extremely long documents. In addition, because soft attention considers the possibility of attending to every entry in the memory at every output timestep, it must wait until the input sequence has been processed before producing output. This makes it inapplicable to real-time sequence transduction problems. Raffel et al. (2017) recently pointed out that these issues can be mitigated when the input-output alignment is *monotonic*, i.e. the correspondence between elements in the input and output sequence does not involve reordering. This property is present in various real-world problems, such as speech recognition and synthesis, where the input and output share a natural temporal order (see, for example, fig. 2). In other settings, the alignment only involves local reorderings, e.g. machine translation for certain language pairs (Birch et al., 2008).

Based on this observation, Raffel et al. (2017) introduced an attention mechanism that explicitly enforces a hard monotonic input-output alignment, which allows for online and linear-time decoding.

---

[*]Equal contribution.

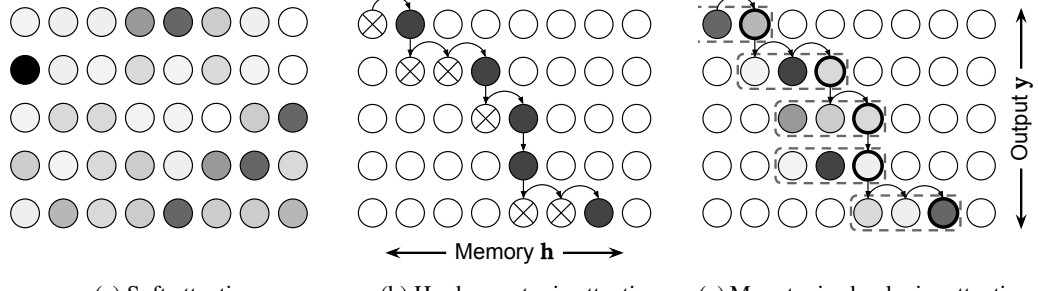

(a) Soft attention.   (b) Hard monotonic attention.   (c) Monotonic chunkwise attention.

Figure 1: Schematics of the attention mechanisms discussed in this paper. Each node represents the possibility of the model attending to a given memory entry (horizontal axis) at a given output timestep (vertical axis). (a) In soft attention, the model assigns a probability (represented by the shade of gray of each node) to each memory entry at each output timestep. The context vector is computed as the weighted average of the memory, weighted by these probabilities. (b) At test time, monotonic attention inspects memory entries from left-to-right, choosing whether to move on to the next memory entry (shown as nodes with ×) or stop and attend (shown as black nodes). The context vector is hard-assigned to the memory entry that was attended to. At the next output timestep, it starts again from where it left off. (c) MoChA utilizes a hard monotonic attention mechanism to choose the endpoint (shown as nodes with bold borders) of the chunk over which it attends. The chunk boundaries (here, with a window size of 3) are shown as dotted lines. The model then performs soft attention (with attention weighting shown as the shade of gray) over the chunk, and computes the context vector as the chunk's weighted average.

However, the hard monotonicity constraint also limits the expressivity of the model compared to soft attention (which can induce an arbitrary soft alignment). Indeed, experimentally it was shown that the performance of sequence-to-sequence models utilizing this monotonic attention mechanism lagged behind that of standard soft attention.

In this paper, we aim to close this gap by introducing a novel attention mechanism which retains the online and linear-time benefits of hard monotonic attention while allowing for soft alignments. Our approach, which we dub "**Mo**notonic **Ch**unkwise **A**ttention" (MoChA), allows the model to perform soft attention over small chunks of the memory preceding where a hard monotonic attention mechanism has chosen to attend. It also has a training procedure which allows it to be straightforwardly applied to existing sequence-to-sequence models and trained with standard backpropagation. We show experimentally that MoChA effectively closes the gap between monotonic and soft attention on online speech recognition and provides a 20% relative improvement over monotonic attention on document summarization (a task which does not exhibit monotonic alignments). These benefits incur only a modest increase in the number of parameters and computational cost. We also provide a discussion of related work and ideas for future research using our proposed mechanism.

## 2  DEFINING MoChA

To develop our proposed attention mechanism, we will first review the sequence-to-sequence framework and the most common form of soft attention used with it. Because MoChA can be considered a generalization of monotonic attention, we then re-derive this approach and point out some of its shortcomings. From there, we show how soft attention over chunks can be straightforwardly added to hard monotonic attention, giving us the MoChA attention mechanism. We also show how MoChA can be trained efficiently with respect to the mechanism's expected output, which allows us to use standard backpropagation.

### 2.1  SEQUENCE-TO-SEQUENCE MODELS

A sequence-to-sequence model is one which transduces an input sequence $\mathbf{x} = \{x_1, \ldots, x_T\}$ to an output sequence (potentially of a different modality) $\mathbf{y} = \{y_1, \ldots, y_U\}$. Typically, the input sequence is first converted to a sequence of hidden states $\mathbf{h} = \{h_1, \ldots, h_T\}$ by an encoder recurrent

neural network (RNN):

$$h_j = \text{EncoderRNN}(x_j, h_{j-1}) \tag{1}$$

A decoder RNN then updates its hidden state autoregressively and an output layer (typically using a $\text{softmax}$ nonlinearity) produces the output sequence:

$$s_i = \text{DecoderRNN}(y_{i-1}, s_{i-1}, c_i) \tag{2}$$
$$y_i = \text{Output}(s_i, c_i) \tag{3}$$

where $s_i$ is the decoder's state and $c_i$ is a "context" vector which is computed as a function of the encoder hidden state sequence $\mathbf{h}$. Note that $c_i$ is the sole conduit through which the decoder has access to information about the input sequence.

In the originally proposed sequence-to-sequence framework (Sutskever et al., 2014), the context vector is simply set to the final encoder hidden state, i.e. $c_i = h_T$. It was subsequently found that this approach exhibits degraded performance when transducing long sequences (Bahdanau et al., 2015). Instead, it has become standard to use an *attention mechanism* which treats the hidden state sequence as a (soft-)addressable memory whose entries are used to compute the context vector $c_i$. In the following subsections, we discuss three such approaches for computing $c_i$; otherwise, the sequence-to-sequence framework remains unchanged.

## 2.2    STANDARD SOFT ATTENTION

Currently, the most commonly used attention mechanism is the one originally proposed in (Bahdanau et al., 2015). At each output timestep $i$, this approach proceeds as follows: First, an unnormalized scalar "energy" value $e_{i,j}$ is produced for each memory entry:

$$e_{i,j} = \text{Energy}(h_j, s_{i-1}) \tag{4}$$

A common choice for $\text{Energy}(\cdot)$ is

$$\text{Energy}(h_j, s_{i-1}) := v^\top \tanh(W_h h_j + W_s s_{i-1} + b) \tag{5}$$

where $W_h \in \mathbb{R}^{d \times \dim(h_j)}$, $W_s \in \mathbb{R}^{d \times \dim(s_{i-1})}$, $b \in \mathbb{R}^d$ and $v \in \mathbb{R}^d$ are learnable parameters and $d$ is the hidden dimensionality of the energy function. Second, these energy scalars are normalized across the memory using the $\text{softmax}$ function to produce weighting values $\alpha_{i,j}$:

$$\alpha_{i,j} = \frac{\exp(e_{i,j})}{\sum_{k=1}^{T} \exp(e_{i,k})} = \text{softmax}(e_{i,:})_j \tag{6}$$

Finally, the context vector is computed as a simple weighted average of $\mathbf{h}$, weighted by $\alpha_{i,:}$:

$$c_i = \sum_{j=1}^{T} \alpha_{i,j} h_j \tag{7}$$

We visualize this soft attention mechanism in fig. 1a.

Note that in order to compute $c_i$ for any output timestep $i$, we need to have computed all of the encoder hidden states $h_j$ for $j \in \{1, \ldots, T\}$. This implies that this form of attention is not applicable to online/real-time sequence transduction problems, because it needs to have observed the entire input sequence before producing any output. Furthermore, producing each context vector $c_i$ involves computing $T$ energy scalar terms and weighting values. While these operations can typically be parallelized, this nevertheless results in decoding having a $\mathcal{O}(TU)$ cost in time and space.

## 2.3    MONOTONIC ATTENTION

To address the aforementioned issues with soft attention, Raffel et al. (2017) proposed a hard monotonic attention mechanism whose attention process can be described as follows: At output timestep $i$, the attention mechanism begins inspecting memory entries starting at the memory index it attended to at the previous output timestep, referred to as $t_{i-1}$. It then computes an unnormalized energy scalar $e_{i,j}$ for $j = t_{i-1}, t_{i-1} + 1, \ldots$ and passes these energy values into a logistic sigmoid

function $\sigma(\cdot)$ to produce "selection probabilities" $p_{i,j}$. Then, a discrete attend/don't attend decision $z_{i,j}$ is sampled from a Bernoulli random variable parameterized by $p_{i,j}$. In total, so far we have

$$e_{i,j} = \text{MonotonicEnergy}(s_{i-1}, h_j) \tag{8}$$
$$p_{i,j} = \sigma(e_{i,j}) \tag{9}$$
$$z_{i,j} \sim \text{Bernoulli}(p_{i,j}) \tag{10}$$

As soon as $z_{i,j} = 1$ for some $j$, the model stops and sets $t_i = j$ and $c_i = h_{t_i}$. This process is visualized in fig. 1b. Note that because this attention mechanism only makes a single pass over the memory, it has a $\mathcal{O}(\max(T, U))$ (linear) cost. Further, in order to attend to memory entry $h_j$, the encoder RNN only needs to have processed input sequence entries $x_1, \ldots, x_j$, which allows it to be used for online sequence transduction. Finally, note that if $p_{i,j} \in \{0, 1\}$ (a condition which is encouraged, as discussed below) then the greedy assignment of $c_i = h_{t_i}$ is equivalent to marginalizing over possible alignment paths.

Because this attention process involves sampling and hard assignment, models utilizing hard monotonic attention can't be trained with backpropagation. To remedy this, Raffel et al. (2017) propose training with respect to the expected value of $c_i$ by computing the probability distribution over the memory induced by the attention process. This distribution takes the following form:

$$\alpha_{i,j} = p_{i,j} \left( (1 - p_{i,j-1}) \frac{\alpha_{i,j-1}}{p_{i,j-1}} + \alpha_{i-1,j} \right) \tag{11}$$

The context vector $c_i$ is then computed as a weighted sum of the memory as in eq. (7). Equation (11) can be explained by observing that $(1 - p_{i,j-1})\alpha_{i,j-1}/p_{i,j-1}$ is the probability of attending to memory entry $j - 1$ at the current output timestep ($\alpha_{i,j-1}$) corrected for the fact that the model did not attend to memory entry $j$ (by multiplying by $(1 - p_{i,j-1})$ and dividing by $p_{i,j-1}$). The addition of $\alpha_{i-1,j}$ represents the additional possibility that the model attended to entry $j$ at the previous output timestep, and finally multiplying it all by $p_{i,j}$ reflects the probability that the model selected memory item $j$ at the current output timestep $i$. Note that this recurrence relation is not parallelizable across memory indices $j$ (unlike, say, $\text{softmax}$), but fortunately substituting $q_{i,j} = \alpha_{i,j}/p_{i,j}$ produces the first-order linear difference equation $q_{i,j} = (1 - p_{i,j-1})q_{i,j-1} + \alpha_{i-1,j}$ which has the following solution (Kelley & Peterson, 2001):

$$q_{i,:} = \text{cumprod}(1 - p_{i,:}) \, \text{cumsum} \left( \frac{\alpha_{i-1,:}}{\text{cumprod}(1 - p_{i,:})} \right) \tag{12}$$

where $\text{cumprod}(\mathbf{x}) = [1, x_1, x_1 x_2, \ldots, \prod_i^{|x|-1} x_i]$ and $\text{cumsum}(\mathbf{x}) = [x_1, x_1 + x_2, \ldots, \sum_i^{|x|} x_i]$. Because the cumulative sum and product can be computed in parallel (Ladner & Fischer, 1980), models can still be trained efficiently with this approach.

Note that training is no longer online or linear-time, but the proposed solution is to use this "soft" monotonic attention for training and use the hard monotonic attention process at test time. To encourage discreteness, Raffel et al. (2017) used the common approach of adding zero-mean, unit-variance Gaussian noise to the logistic sigmoid function's activations, which causes the model to learn to produce effectively binary $p_{i,j}$. If $p_{i,j}$ are binary, $z_{i,j} = \mathbb{1}(p_{i,j} > .5)$, so in practice sampling is eschewed at test-time in favor of simple thresholding. Separately, it was observed that switching from the $\text{softmax}$ nonlinearity to the logistic sigmoid resulted in optimization issues due to saturating and sensitivity to offset. To mitigate this, a slightly modified energy function was used:

$$\text{MonotonicEnergy}(s_{i-1}, h_j) = g \frac{v^\top}{\|v\|} \tanh(W_s s_{i-1} + W_h h_j + b) + r \tag{13}$$

where $g, r$ are learnable scalars and $v, W_s, W_h, b$ are as in eq. (5). Further discussion of these modifications is provided in (Raffel et al., 2017) appendix G.

## 2.4 MONOTONIC CHUNKWISE ATTENTION

While hard monotonic attention provides online and linear-time decoding, it nevertheless imposes two significant constraints on the model: First, that the decoder can only attend to a single entry in memory at each output timestep, and second, that the input-output alignment must be *strictly* monotonic. These constraints are in contrast to standard soft attention, which allows a potentially

---

**Algorithm 1** MoChA decoding process (test time). During training, lines 4-19 are replaced with eqs. (20) to (26) and $y_{i-1}$ is replaced with the ground-truth output at timestep $i-1$.

---

1: **Input:** memory $\mathbf{h}$ of length $T$, chunk size $w$
2: **State:** $s_0 = \vec{0}, t_0 = 1, i = 1, y_0 = \text{StartOfSequence}$
3: **while** $y_{i-1} \neq \text{EndOfSequence}$ **do**    // *Produce output tokens until end-of-sequence token is produced*
4:     **for** $j = t_{i-1}$ **to** $T$ **do**    // *Start inspecting memory entries $h_j$ left-to-right from where we left off*
5:         $e_{i,j} = \text{MonotonicEnergy}(s_{i-1}, h_j)$    // *Compute attention energy for $h_j$*
6:         $p_{i,j} = \sigma(e_{i,j})$    // *Compute probability of choosing $h_j$*
7:         **if** $p_{i,j} \geq 0.5$ **then**    // *If $p_{i,j}$ is larger than 0.5, we stop scanning the memory*
8:             $v = j - w + 1$    // *Set chunk start location*
9:             **for** $k = v$ **to** $j$ **do**    // *Compute chunkwise* softmax *energies over a size-w chunk before $j$*
10:                 $u_{i,k} = \text{ChunkEnergy}(s_{i-1}, h_k)$
11:             **end for**
12:             $c_i = \sum_{k=v}^{j} \frac{\exp(u_{i,k})}{\sum_{l=v}^{j} \exp(u_{i,l})} h_k$    // *Compute* softmax-*weighted average over the chunk*
13:             $t_i = j$    // *Remember where we left off for the next output timestep*
14:             **break**    // *Stop scanning the memory*
15:         **end if**
16:     **end for**
17:     **if** $p_{i,j} < 0.5, \forall j \in \{t_{i-1}, t_{i-1}+1, \ldots, T\}$ **then**
18:         $c_i = \vec{0}$    // *If we scanned the entire memory without stopping, set $c_i$ to a vector of zeros*
19:     **end if**
20:     $s_i = \text{DecoderRNN}(s_{i-1}, y_{i-1}, c_i)$    // *Update output RNN state based on the new context vector*
21:     $y_i = \text{Output}(s_i, c_i)$    // *Output a new symbol using the* softmax *output layer*
22:     $i = i + 1$
23: **end while**

---

arbitrary and smooth input-output alignment. Experimentally, it was shown that performance degrades somewhat on all tasks tested in (Raffel et al., 2017). Our hypothesis is that this degradation stems from the aforementioned constraints imposed by hard monotonic attention.

To remedy these issues, we propose a novel attention mechanism which we call MoChA, for **Mo**notonic **Ch**unkwise **A**ttention. The core of our idea is to allow the attention mechanism to perform soft attention over small "chunks" of memory preceding where a hard monotonic attention mechanism decides to stop. This facilitates some degree of softness in the input-output alignment, while retaining the online decoding and linear-time complexity benefits.

At test time, we follow the hard monotonic attention process of section 2.3 in order to determine $t_i$ (the location where the hard monotonic attention mechanism decides to stop scanning the memory at output timestep $i$). However, instead of setting $c_i = h_{t_i}$, we allow the model to perform soft attention over the length-$w$ window of memory entries preceding and including $t_i$:

$$v = t_i - w + 1 \tag{14}$$

$$u_{i,k} = \text{ChunkEnergy}(s_{i-1}, h_k), k \in \{v, v+1, \ldots, t_i\} \tag{15}$$

$$c_i = \sum_{k=v}^{t_i} \frac{\exp(u_{i,k})}{\sum_{l=v}^{t_i} \exp(u_{i,l})} h_k \tag{16}$$

where $\text{ChunkEnergy}(\cdot)$ is an energy function analogous to eq. (5), which is distinct from the $\text{MonotonicEnergy}(\cdot)$ function. MoChA's attention process is visualized in fig. 1c. Note that MoChA allows for nonmonotonic alignments; specifically, it allows for reordering of the memory entries $h_v, \ldots, h_{t_i}$. Including soft attention over chunks only increases the runtime complexity by the constant factor $w$, and decoding can still proceed in an online fashion. Furthermore, using MoChA only incurs a modest increase in the total number of parameters (corresponding to adding the second attention energy function $\text{ChunkEnergy}(\cdot)$). For example, in the speech recognition experiments described in section 3.1, the total number of model parameters only increased by about 1%. Finally, we point out that setting $w = 1$ recovers hard monotonic attention. For completeness, we show the decoding algorithm for MoChA in full in algorithm 1.

During training, we proceed in a similar fashion as with monotonic attention, namely training the model using the expected value of $c_i$ based on MoChA's induced probability distribution (which we

denote $\beta_{i,j}$). This can be computed as

$$\beta_{i,j} = \sum_{k=j}^{j+w-1} \left( \alpha_{i,k} \exp(u_{i,j}) \bigg/ \sum_{l=k-w+1}^{k} \exp(u_{i,l}) \right) \tag{17}$$

The sum over $k$ reflects the possible positions at which the monotonic attention could have stopped scanning the memory in order to contribute probability to $\beta_{i,j}$ and the term inside the summation represents the $\mathrm{softmax}$ probability distribution over the chunk, scaled by the monotonic attention probability $\alpha_{i,k}$. Computing each $\beta_{i,j}$ in this fashion is expensive due to the nested summation. Fortunately, there is an efficient way to compute $\beta_{i,j}$ for $j \in \{1, \ldots, T\}$ in parallel: First, for a sequence $\mathbf{x} = \{x_1, \ldots, x_T\}$ we define

$$\mathrm{MovingSum}(\mathbf{x}, b, f)_n := \sum_{m=n-(b-1)}^{n+f-1} x_m \tag{18}$$

This function can be computed efficiently, for example, by convolving $\mathbf{x}$ with a length-$(f + b - 1)$ sequence of 1s and truncating appropriately. Now, we can compute $\beta_{i,:}$ efficiently as

$$\beta_{i,:} = \exp(u_{i,:}) \, \mathrm{MovingSum} \left( \frac{\alpha_{i,:}}{\mathrm{MovingSum}(\exp(u_{i,:}), w, 1)}, 1, w \right) \tag{19}$$

Putting it all together produces the following algorithm for computing $c_i$ during training:

$$e_{i,j} = \mathrm{MonotonicEnergy}(s_{i-1}, h_j) \tag{20}$$

$$\epsilon \sim \mathcal{N}(0, 1) \tag{21}$$

$$p_{i,j} = \sigma(e_{i,j} + \epsilon) \tag{22}$$

$$\alpha_{i,:} = p_{i,:} \, \mathtt{cumprod}(1 - p_{i,:}) \, \mathtt{cumsum} \left( \frac{\alpha_{i-1,:}}{\mathtt{cumprod}(1 - p_{i,:})} \right) \tag{23}$$

$$u_{i,j} = \mathrm{ChunkEnergy}(s_{i-1}, h_j) \tag{24}$$

$$\beta_{i,:} = \exp(u_{i,:}) \, \mathrm{MovingSum} \left( \frac{\alpha_{i,:}}{\mathrm{MovingSum}(\exp(u_{i,:}), w, 1)}, 1, w \right) \tag{25}$$

$$c_i = \sum_{j=1}^{T} \beta_{i,j} h_j \tag{26}$$

Equations (20) to (23) reflect the (unchanged) computation of the monotonic attention probability distribution, eqs. (24) and (25) compute MoChA's probability distribution, and finally eq. (26) computes the expected value of the context vector $c_i$. In summary, we have developed a novel attention mechanism which allows computing soft attention over small chunks of the memory, whose locations are set adaptively. This mechanism has an efficient training-time algorithm and enjoys online and linear-time decoding at test time. We attempt to quantify the resulting speedup compared to soft attention with a synthetic benchmark in appendix B.

## 3 EXPERIMENTS

To test out MoChA, we applied it to two exemplary sequence transduction tasks: Online speech recognition and document summarization. Speech recognition is a promising setting for MoChA because it induces a naturally monotonic input-output alignment, and because online decoding is often required in the real world. Document summarization, on the other hand, does not exhibit a monotonic alignment, and we mostly include it as a way of testing the limitations of our model. We emphasize that in all experiments, we took a strong baseline sequence-to-sequence model with standard soft attention and changed **only** the attention mechanism; all hyperparameters, model structure, training approach, etc. were kept exactly the same. This allows us to isolate the effective difference in performance caused by switching to MoChA. Of course, this may be an artificially low estimate of the best-case performance of MoChA, due to the fact that it may benefit from a somewhat different hyperparameter setting. We leave eking out the best-case performance for future work.

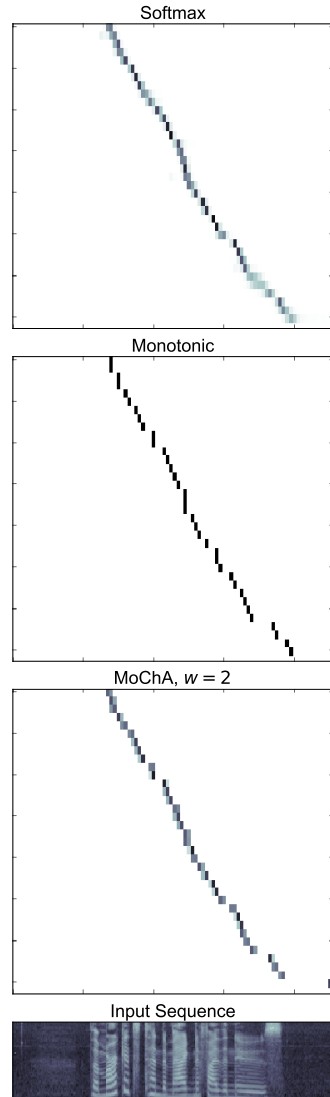

Figure 2: Attention alignments plots and speech utterance feature sequence for the speech recognition task.

Specifically, for MoChA we used eq. (13) for both the MonotonicEnergy and the ChunkEnergy functions. Following (Raffel et al., 2017), we initialized $g = 1/\sqrt{d}$ ($d$ being the attention energy function hidden dimension) and tuned initial values for $r$ based on validation set performance, using $r = -4$ for MoChA on speech recognition, $r = 0$ for MoChA on summarization, and $r = -1$ for our monotonic attention baseline on summarization. We similarly tuned the chunk size $w$: For speech recognition, we were surprised to find that all of $w \in \{2, 3, 4, 6, 8\}$ performed comparably and thus chose the smallest value of $w = 2$. For summarization, we found $w = 8$ to work best. We demonstrate empirically that even these small window sizes give a significant boost over hard monotonic attention ($w = 1$) while incurring only a minor computational penalty. In all experiments, we report metrics on the test set at the training step of best performance on a validation set.

### 3.1 ONLINE SPEECH RECOGNITION

First, we apply MoChA in its natural setting, i.e. a domain where we expect roughly monotonic alignments:[1] Online speech recognition on the Wall Street Journal (WSJ) corpus (Paul & Baker, 1992). The goal in this task is to produce the sequence of words spoken in a recorded speech utterance. In this setting, RNN-based models must be unidirectional in order to satisfy the online requirement. We use the model of (Raffel et al., 2017), which is itself based on that of (Zhang et al., 2016). Full model and training details are provided in appendix A.1, but as a broad overview, the network ingests the spoken utterance as a mel-filterbank spectrogram which is passed to an encoder consisting of convolution layers, convolutional LSTM layers, and unidirectional LSTM layers. The decoder is a single unidirectional LSTM, which attends to the encoder state sequence via either MoChA or a standard soft attention mechanism. The decoder produces a sequence of distributions over character and word-delimiter tokens. Performance is measured in terms of word error rate (WER) after segmenting characters output by the model into words based on the produced word delimiter tokens. None of the models we report integrated a separate language model.

We show the results of our experiments, along with those obtained by prior work, in table 1. MoChA was able to beat the state-of-the-art by a large margin (20% relative). Because the performance of MoChA and the soft attention baseline was so close, we ran 8 repeat trials for both attention mechanisms and report the best, average, and standard deviation of word error rates across these trials. We found MoChA-based models to have slightly higher variance across trials, which resulted in it having a lower best WER but a slightly higher mean WER compared to soft attention (though the difference in means was not statistically significant for $N = 8$ under an unpaired Student's t-test). This is the first time, to our knowledge, that an online attention mechanism matched the performance of standard (offline) soft attention. To get an idea of the behavior of the different attention mechanisms, we show attention alignments for an example from the WSJ validation set in fig. 2. As expected, the alignment looks roughly the same for all attention mechanisms. We note especially that MoChA is indeed taking advantage of the opportunity to produce a soft attention distribution over each length-2 chunk.

Since we empirically found the small value of $w = 2$ to be sufficient to realize these gains, we carried out a few additional experiments to confirm that they can indeed be attributed to MoChA. First, the

---

[1] Even for nonphonemic utterances (e.g. "AAA" being transcribed as "triple A"), the learned alignment still tends to be monotonic – see e.g. (Chan et al., 2016) figure 6.

| Prior Result | WER |
|---|---|
| (Raffel et al., 2017) (CTC baseline) | 33.4% |
| (Luo et al., 2016) (Reinforcement Learning) | 27.0% |
| (Wang et al., 2016) (CTC) | 22.7% |
| (Raffel et al., 2017) (Monotonic Attention) | 17.4% |

| Attention Mechanism | Best WER | Average WER |
|---|---|---|
| Soft Attention (offline) | 14.2% | $14.6 \pm 0.3\%$ |
| MoChA, $w = 2$ | 13.9% | $15.0 \pm 0.6\%$ |

Table 1: Word error rate on the Wall Street Journal test set. Our results (bottom) reflect the statistics of 8 trials.

| Mechanism | R-1 | R-2 |
|---|---|---|
| Soft Attention (offline) | 39.11 | 15.76 |
| Hard Monotonic Attention | 31.14 | 11.16 |
| MoChA, $w = 8$ | 35.46 | 13.55 |

Table 2: ROUGE F-scores for document summarization on the CNN/Daily Mail dataset. The soft attention baseline is our reimplementation of (See et al., 2017).

use of a second independent attention energy function $\mathrm{ChunkEnergy}(\cdot)$ incurs a modest increase in parameter count – about 1% in our speech recognition model. To ensure the improved performance was not due to this parameter increase, we also re-trained the monotonic attention baseline with an energy function with a doubled hidden dimensionality (which produces a comparable increase in the number of parameters in a natural way). Across eight trials, the difference in performance (a decrease of 0.3% WER) was not significant compared to the baseline and was dwarfed by the gains achieved by MoChA. We also trained the $w = 2$ MoChA model with half the attention energy hidden dimensionality (which similarly reconciles the parameter difference) and found it did not significantly undercut our gains, increasing the WER by only 0.2% (not significant over eight trials). Separately, one possible benefit of MoChA is that the attention mechanism can access a larger window of the input when producing the context vectors. An alternative approach towards this end would be to increase the temporal receptive field of the convolutional front-end, so we also re-trained the monotonic attention baseline with this change. Again, the difference in performance (an increase of 0.3% WER) was not significant over eight trials. These additional experiments reinforce the benefits of using MoChA for online speech recognition.

## 3.2 DOCUMENT SUMMARIZATION

Having proven the effectiveness of MoChA in the comfortable setting of speech recognition, we now test its limits in a task without a monotonic input/output alignment. Raffel et al. (2017) experimented with sentence summarization on the Gigaword dataset, which frequently exhibits monotonic alignments and involves short sequences (sentence-length sequences of words). They were able to achieve only slightly degraded performance with hard monotonic attention compared to a soft attention baseline. As a result, we turn to a more difficult task where hard monotonic attention struggles more substantially due to the lack of monotonic alignments: Document summarization on the CNN/Daily Mail corpus (Nallapati et al., 2016). While we primarily study this problem because it has the potential to be challenging, online and linear-time attention could also be beneficial in real-world scenarios where very long bodies of text need to be summarized as they are being created (e.g. producing a summary of a speech as it is being given).

The goal of this task is to produce a sequence of "highlight" sentences from a news article. As a baseline model, we chose the "pointer-generator" network (without the coverage penalty) of (See et al., 2017). For full model architecture and training details, refer to appendix A.2. As a brief summary, input words are converted to a learned embedding and passed into the model's encoder, consisting of a single bidirectional LSTM layer. The decoder is a unidirectional LSTM with an attention mechanism whose state is passed to a $\mathrm{softmax}$ layer which produces a sequence of distributions over the vocabulary. The model is augmented with a copy mechanism, which interpolates linearly between using the $\mathrm{softmax}$ output layer's word distribution, or a distribution of word IDs weighted by the attention distribution at a given output timestep. We tested this model with standard soft attention (as used in (See et al., 2017)), hard monotonic attention, and MoChA with $w = 8$.

The results are shown in table 2. We found that using a hard monotonic attention mechanism degraded performance substantially (nearly 8 ROUGE-1 points), likely because of the strong reordering required by this task. However, MoChA was able to effectively halve the gap between monotonic and soft attention, despite using the modest chunk size of $w = 8$. We consider this an encouraging indication of the benefits of being able to deal with local reorderings.

## 4 RELATED WORK

A similar model to MoChA is the "Neural Transducer" (Jaitly et al., 2015), where the input sequence is pre-segmented into equally-sized non-overlapping chunks and attentive sequence-to-sequence transduction is performed over each chunk separately. The full output sequence is produced by marginalizing out over possible end-of-sequence locations for the sequences generated from each chunk. While our model also performs soft attention over chunks, the locations of our chunks are set adaptively by a hard monotonic attention mechanism rather than fixed, and it avoids the marginalization over chunkwise end-of-sequence tokens.

Chorowski et al. (2015) proposes a similar idea, wherein the range over which soft attention is computed at each output timestep is limited to a fixed-sized window around the memory index of maximal attention probability at the previous output timestep. While this also produces soft attention over chunks, our approach differs in that the chunk boundary is set by an independent hard monotonic attention mechanism. This difference resulted in Chorowski et al. (2015) using a very large chunk size of 150, which effectively prevents its use in online settings and incurs a significantly higher computational cost than our approach which only required small values for $w$.

A related class of non-attentive sequence transduction models which can be used in online settings are connectionist temporal classification (Graves et al., 2006), the RNN transducer (Graves, 2012), segment-to-segment neural transduction (Yu et al., 2016), and the segmental RNN (Kong et al., 2015). These models are distinguished from sequence-to-sequence models with attention mechanisms by the fact that the decoder does not condition directly on the input sequence, and that decoding is done via a dynamic program. A detailed comparison of this class of approaches and attention-based models is provided in (Prabhavalkar et al., 2017), where it is shown that attention-based models perform best in speech recognition experiments. Further, Hori et al. (2017) recently proposed jointly training a speech recognition model with both a CTC loss and an attention mechanism. This combination encouraged the model to learn monotonic alignments, but Hori et al. (2017) still used a standard soft attention mechanism which precludes the model's use in online settings.

Finally, we note that there have been a few other works considering hard monotonic alignments, e.g. using reinforcement learning (Zaremba & Sutskever, 2015; Luo et al., 2016; Lawson et al., 2017), by using separately-computed target alignments (Aharoni & Goldberg, 2016) or by assuming a strictly diagonal alignment (Luong et al., 2015). We suspect that these approaches may confer similar benefits from adding chunkwise attention.

## 5 CONCLUSION

We have proposed MoChA, an attention mechanism which performs soft attention over adaptively-located chunks of the input sequence. MoChA allows for online and linear-time decoding, while also facilitating local input-output reorderings. Experimentally, we showed that MoChA obtains state-of-the-art performance on an online speech recognition task, and that it substantially outperformed a hard monotonic attention-based model on document summarization. In future work, we are interested in applying MoChA to additional problems with (approximately) monotonic alignments, such as speech synthesis (Wang et al., 2017) and morphological inflection (Aharoni & Goldberg, 2016). We would also like to investigate ways to allow the chunk size $w$ to also vary adaptively. To facilitate building on our work, we provide an example implementation of MoChA online.[2]

### ACKNOWLEDGMENTS

We thank Ying Xiao, Kevin Clark, Jacob Buckman, our anonymous reviewers, and members of the Google Brain Team for their helpful comments on this paper.

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

# A   EXPERIMENT DETAILS

In this appendix, we provide specifics about the experiments carried out in section 3. All experiments were done using TensorFlow (Abadi et al., 2016).

## A.1   ONLINE SPEECH RECOGNITION

Overall, our model follows that of (Raffel et al., 2017), but we repeat the details here for posterity. We represented speech utterances as mel-scaled spectrograms with 80 coefficients, along with delta and delta-delta coefficients. Feature sequences were first fed into two convolutional layers, each with $3 \times 3$ filters and a $2 \times 2$ stride with 32 filters per layer. Each convolution was followed by batch normalization (Ioffe & Szegedy, 2015) prior to a ReLU nonlinearity. The output of the convolutional layers was fed into a convolutional LSTM layer, using $1 \times 3$ filters. This was followed by an additional $3 \times 3$ convolutional layer with 32 filters and a stride of $1 \times 1$. Finally, the encoder had three additional unidirectional LSTM layers with a hidden state size of 256, each followed by a dense layer with a 256-dimensional output with batch normalization and a ReLU nonlinearity.

The decoder was a single unidirectional LSTM layer with a hidden state size of 256. Its input consisted of a 64-dimensional learned embedding of the previously output symbol and the 256-dimensional context vector produced by the attention mechanism. The attention energy function had a hidden dimensionality $d$ of 128. The $\mathrm{softmax}$ output layer took as input the concatenation of the attention context vector and the decoder's state.

The network was trained using the Adam optimizer (Kingma & Ba, 2014) with $\beta_1 = 0.9$, $\beta_2 = 0.999$, and $\epsilon = 10^{-6}$. The initial learning rate $0.001$ was dropped by a factor of $10$ after 600,000, 800,000, and 1,000,000 steps. Note that Raffel et al. (2017) used a slightly different learning rate schedule, but we found that the aforementioned schedule improved performance both for the soft attention baseline and for MoChA, but hurt performance for hard monotonic attention. For this reason, we report the hard monotonic attention performance from (Raffel et al., 2017) instead of re-running that baseline. Inputs were fed into the network in batches of 8 utterances, using standard teacher forcing. Localized label smoothing (Chorowski & Jaitly, 2017) was applied to the target outputs with weights $[0.015, 0.035, 0.035, 0.015]$ for neighbors at $[-2, -1, 1, 2]$. We used gradient clipping, setting the norm of the global gradient vector to 1 whenever it exceeded that threshold. We added variational weight noise to LSTM layer parameters and embeddings with standard deviation of $0.075$ starting after 20,000 training steps. We also applied L2 weight decay with a coefficient of $10^{-6}$. At test time, we used a beam search with rank pruning at 8 hypotheses and a pruning threshold of 3.

## A.2   DOCUMENT SUMMARIZATION

For summarization, we reimplemented the pointer-generator of See et al. (2017). Inputs were provided as one-hot vectors representing ID in a 50,000 word vocabulary, which were mapped to a 512-dimensional learned embedding. The encoder consisted of a single bidirectional LSTM layer with 512 hidden units, and the decoder was a single unidirectional LSTM layer with 1024 hidden units. Our attention mechanisms had a hidden dimensionality $d$ of 1024. Output words were embedded into a learned 1024-dimensional embedding and concatenated with the context vector before being fed back in to the decoder.

For training, we used the Adam optimizer with $\beta_1 = 0.9$, $\beta_2 = 0.999$, and $\epsilon = 0.0000008$. Our optimizer had an initial learning rate of $0.0005$ which was continuously decayed starting at 50,000 steps such that the learning rate was halved every 10,000 steps until it reached $0.00005$. Sequences were fed into the model with a batch size of 64. As in See et al. (2017), we truncated all input sequence to a maximum length of $400$ words. The global norm of the gradient was clipped to never exceed 5. Note that we did not include the "coverage penalty" discussed in See et al. (2017) in our models. During eval, we used an identical beam search as in the speech recognition experiments with rank pruning at 8 hypotheses and a pruning threshold of 3.

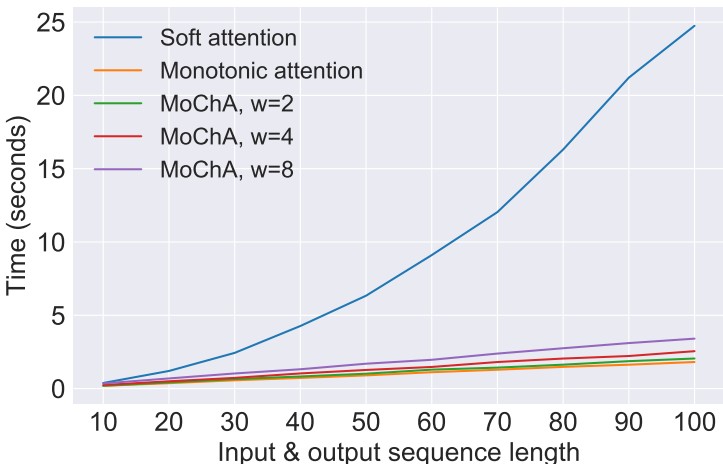

Figure 3: Speeds of different attention mechanisms on a synthetic benchmark.

## B  SPEED BENCHMARK

To get an idea of the possible speedup incurred by using MoChA instead of standard soft attention, we carried out a simple synthetic benchmark analogous to the one in (Raffel et al., 2017), appendix F. In this test, we implemented *solely* the attention mechanism and measured its speed for various input/output sequence lengths. This isolates the speed of the portion of the network we are studying; in practice, other portions of the network (e.g. encoder RNN, decoder RNN, etc.) may dominate the computational cost of running the full model. Any resulting speedup can therefore be considered an upper bound on what might be observed in the real-world. Further, we coded the benchmark in C++ using the Eigen library (Guennebaud et al., 2010) to remove any overhead incurred by a particular model framework.

In this synthetic setting, attention was performed over a randomly generated encoder hidden state sequence, using random decoder states. The encoder and decoder state dimensionality was set to 256. We varied the input and output sequence lengths $T$ and $U$ simultaneously, in the range $\{10, 20, 30, \ldots, 100\}$. We measured the speed for soft attention, monotonic attention (i.e. MoChA with $w = 1$), and MoChA with $w = \{2, 4, 8\}$. For all times, we report the mean of 100 trials.

The results are shown in fig. 3. As expected, soft attention exhibits a roughly quadratic time complexity, where as MoChA's is linear. This results in a larger speedup factor as $T$ and $U$ increase. Further, the complexity of MoChA increases linearly with $w$. Finally, note that for $T, U = 10$ and $w = 8$, the speed of MoChA and soft attention are similar, because the chunk effectively spans the entire memory. This confirms the intuition that speedups from MoChA will be most dramatic for large $T$ and $U$ and relatively small $w$.

## C  MONOTONIC ADAPTIVE CHUNKWISE ATTENTION (MATCHA)

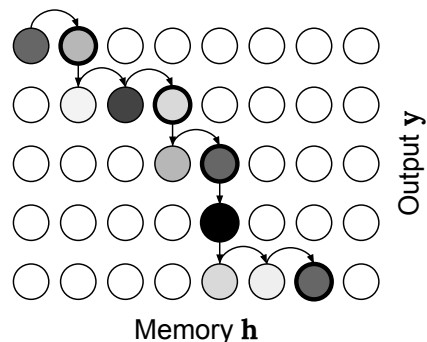

Figure 4: Schematic of the test-time decoding procedure of MAtChA. The semantics of the nodes and horizontal and vertical axes are as in figs. 1a to 1c. MAtChA performs soft attention over variable-sized chunks set by the locations attended to by a monotonic attention mechanism.

In this paper, we considered an attention mechanism which attends to small, fixed-length chunks preceding the location set by a monotonic attention mechanism. In parallel with this work, we also considered another online and linear-time attention mechanism which instead set the chunks to be the region of memory between $t_i$ and $t_{i-1}$. We called this approach MAtChA, for **M**onotonic **A**dap**t**ive **Ch**unkwise **A**ttention. The motivation behind this alternative was that in some cases it may be suboptimal to use a fixed chunk size for all locations in all sequences. However, as we will discuss below, we found that it did not improve performance over MoChA on any of the tasks we tried despite the training-time algorithm having increased memory and computational requirements. We include a discussion and derivation of MAtChA here for posterity, in case other researchers are interested in pursuing similar ideas.

Overall, the test-time decoding process of MAtChA (which is also online and linear-time) is extremely similar to algorithm 1 except that instead of setting the chunk start location $v$ to $v = j - w + 1$, we set $v = t_{i-1}$ so that

$$c_i = \sum_{k=t_{i-1}}^{t_i} \frac{\exp(u_{i,k})}{\sum_{l=t_{i-1}}^{t_i} \exp(u_{i,l})} h_k \qquad (27)$$

This process is visualized in fig. 4. Note that if $t_i = t_{i-1}$, then MAtChA must assign all attention to memory entry $t_i$ because the sole entry of the chunk must be assigned a probability of 1.

The overall equation for MAtChA's attention for memory entry $j$ at output timestep $i$ can be expressed as

$$\beta_{i,j} = \sum_{k=1}^{j} \sum_{l=j}^{T} \left( \frac{\exp(u_{i,j})}{\sum_{m=k}^{l} \exp(u_{i,m})} \alpha_{i-1,k} p_{i,l} \prod_{n=k}^{l-1} (1 - p_{i,n}) \right) \qquad (28)$$

This equation can be explained left to right as follows: First, we must sum over all possible positions that monotonic attention could have attended to at the previous timestep $k \in \{1, \dots, j\}$. Second, we sum over all possible locations where we can attend at the current output timestep $l \in \{j, \dots, T\}$. Third, for a given input/output timestep combination, we compute the softmax probability of memory entry $j$ over the chunk spanning from $k$ to $l$ (as in the main text, we refer to attention energies produced by ChunkEnergy as $u_{i,j}$). Fourth, we multiply by $\alpha_{i-1,k}$ which represents the probability that the monotonic attention mechanism attended to memory entry $k$ at the previous timestep. Fifth, we multiply by $p_{i,l}$, the probability of the monotonic attention mechanism choosing memory entry $l$ at the current output timestep. Finally, we multiply by the probability of *not* choosing any of the memory entries from $k$ to $l - 1$. Using eq. (28) to compute $\beta_{i,j}$ to obtain the expected value of the context vector $c_i$ allows models utilizing MAtChA to be trained with backpropagation.

Note that eq. (28) contains multiple nested summations and products for computing each $i, j$ pair. Fortunately, as with monotonic attention and MoChA there is a dynamic program which allows $\beta_{i,:}$ to be computed completely in parallel which can be derived as follows:

$$\beta_{i,j} = \sum_{k=1}^{j} \sum_{l=j}^{T} \left( \frac{\exp(u_{i,j})}{\sum_{m=k}^{l} \exp(u_{i,m})} \alpha_{i-1,k} p_{i,l} \prod_{n=k}^{l-1} (1 - p_{i,n}) \right) \qquad (29)$$

$$= \exp(u_{i,j}) \sum_{k=1}^{j} \sum_{l=j}^{T} \left( \frac{\alpha_{i-1,k}}{\sum_{m=k}^{l} \exp(u_{i,m})} p_{i,l} \prod_{n=k}^{l-1} (1 - p_{i,n}) \right) \qquad (30)$$

$$= \exp(u_{i,j}) \sum_{l=j}^{T} \sum_{k=1}^{j} \left( \frac{\alpha_{i-1,k}}{\sum_{m=k}^{l} \exp(u_{i,m})} p_{i,l} \prod_{n=k}^{l-1} (1 - p_{i,n}) \right) \qquad (31)$$

$$= \exp(u_{i,j}) \sum_{l=j}^{T} p_{i,l} \sum_{k=1}^{j} \left( \frac{\alpha_{i-1,k}}{\sum_{m=k}^{l} \exp(u_{i,m})} \prod_{n=k}^{l-1} (1 - p_{i,n}) \right) \tag{32}$$

$$= \exp(u_{i,j}) \sum_{l=j}^{T} p_{i,l} \sum_{k=1}^{j} \left( \frac{\alpha_{i-1,k}}{\sum_{m=k}^{l} \exp(u_{i,m})} \prod_{n=k}^{j-1} (1 - p_{i,n}) \prod_{o=j}^{l-1} (1 - p_{i,o}) \right) \tag{33}$$

$$= \exp(u_{i,j}) \sum_{l=j}^{T} p_{i,l} \prod_{o=j}^{l-1} (1 - p_{i,o}) \sum_{k=1}^{j} \left( \frac{\alpha_{i-1,k}}{\sum_{m=k}^{l} \exp(u_{i,m})} \prod_{n=k}^{j-1} (1 - p_{i,n}) \right) \tag{34}$$

$$r_{i,j,l} = \sum_{k=1}^{j} \left( \frac{\alpha_{i-1,k}}{\sum_{m=k}^{l} \exp(u_{i,m})} \prod_{n=k}^{j-1} (1 - p_{i,n}) \right) \tag{35}$$

$$= \sum_{k=1}^{j-1} \left( \frac{\alpha_{i-1,k}}{\sum_{m=k}^{l} \exp(u_{i,m})} \prod_{n=k}^{j-1} (1 - p_{i,n}) \right) + \frac{\alpha_{i-1,j}}{\sum_{m=j}^{l} \exp(u_{i,m})} \tag{36}$$

$$= (1 - p_{i,j-1}) \sum_{k=1}^{j-1} \left( \frac{\alpha_{i-1,k}}{\sum_{m=k}^{l} \exp(u_{i,m})} \prod_{n=k}^{j-2} (1 - p_{i,n}) \right) + \frac{\alpha_{i-1,j}}{\sum_{m=j}^{l} \exp(u_{i,m})} \tag{37}$$

$$= (1 - p_{i,j-1}) r_{i,j-1,l} + \frac{\alpha_{i-1,j}}{\sum_{m=j}^{l} \exp(u_{i,m})} \tag{38}$$

$$\beta_{i,j} = \exp(u_{i,j}) \sum_{l=j}^{T} p_{i,l} \prod_{o=j}^{l-1} (1 - p_{i,o}) r_{i,j,l} \tag{39}$$

Note that eq. (38) has the same form as eq. (11); following the derivation of (Raffel et al., 2017) appendix C.1 suggests that it can similarly be expressed in terms of (parallelizable) cumulative sum and cumulative product operations. However, a notable difference between eq. (38) and eq. (11) is that the former has a dependence on an additional index variable $l$. This is due to the fact that computing $r_{i,j,l}$ for all $j$ and $l$ requires computing the sum of all possible subsequences of $\exp(u_{i,:})$. Fortunately, these subsequence sums can also be computed efficiently; first, define

$$\text{AllPartialSums}(\mathbf{x})_{j,l} = \begin{cases} \sum_{m=j}^{l} x_m, j \leq l \\ 1, j > l \end{cases} \tag{40}$$

Note that, for a sequence $\mathbf{x}$ of length $T$, $\text{AllPartialSums}(\mathbf{x})$ produces a matrix of shape $T \times T$. Now, for $j \leq l$ we have

$$\text{AllPartialSums}(\mathbf{x})_{j,l} = (x_1 + x_2 + \ldots + x_l) - (0 + x_1 + \ldots + x_{j-1}) \tag{41}$$

The sum in the first set of parentheses is simply the $l$th entry of the cumulative sum of $\mathbf{x}$; the sum in the second is the $j$th entry of the *exclusive* cumulative sum of $x$. It follows that all entries of $\text{AllPartialSums}(\mathbf{x})$ can be computed efficiently in parallel by computing the cumulative sum and subtracting appropriately. Combining the above and the derivation of (Raffel et al., 2017) appendix C.1, we have that

$$r_{i,:,:} = \texttt{cumprod}(1 - p_{i,:}) \, \texttt{cumsum} \left( \frac{\alpha_{i-1,:}}{\text{AllPartialSums}(\exp(u_{i,:})) \, \texttt{cumprod}(1 - p_{i,:})} \right) \tag{42}$$

where, as a minor abuse, we are using "broadcasting"[3] notation. In a similar fashion, the product over terms $(1 - p_{i,o})$ in eq. (39) involves computing the product of all possible subsequences. A function $\text{AllPartialProducts}(\cdot)$ can be analogously defined to eq. (40) and computed efficiently with a cumulative product and division. Putting it all together, we can compute all of the terms $\beta_{i,j}$ in parallel for a given output timestep $i$ as

$$\beta_{i,:} = \exp(u_{i,:}) \sum_{l=j}^{T} p_{i,:} \, \text{AllPartialProducts}(1 - p_{i,:})_{:,l} r_{i,:,l} \tag{43}$$

---

[3]https://docs.scipy.org/doc/numpy-1.13.0/user/basics.broadcasting.html

While we have demonstrated a parallelizable procedure for computing MAtChA's attention distribution, the marginalization over all possible chunk start and end locations necessitates a quadratic number of terms to be computed for each output timestep/memory entry combination. Even in the case of perfectly efficient parallelization, the result is an algorithm which requires $\mathcal{O}(UT^2)$ memory for decoding (as opposed to the $\mathcal{O}(UT)$ memory required when training standard soft attention, monotonic attention, or MoChA). This puts it at a distinct disadvantage, especially for large values of $T$. Experimentally, we had hoped that these drawbacks would be outweighed by superior empirical performance of MAtChA, but we unfortunately found that it did not perform any better than MoChA for the tasks we tried. As a result, we decided not to include discussion of MAtChA in the main text and recommend against its use in its current form. Nevertheless, we are interested in mixing MoChA and MAtChA in future work in attempt to reap the benefits of their combined strength.

