# OpenReview forum: "Monotonic Chunkwise Attention"
_ICLR.cc/2018/Conference — Accept (Poster)_

### Official Review · AnonReviewer1 · 2017-11-23
**Great writing but lacks analysis**

**Rating:** 7
**Confidence:** 5

**Review:**

This paper proposes a small modification to the monotonic attention in [1] by adding a soft attention to the segment predicted by the monotonic attention. The paper is very well written and easy to follow. The experiments are also convincing. Here are a few suggestions and questions to make the paper stronger.

The first set of questions is about the monotonic attention. Training the monotonic attention with expected context vectors is intuitive, but can this be justified further? For example, how far does using the expected context vector deviate from marginalizing the monotonic attention? The greedy step, described in the first paragraph of page 4, also has an effect on the produced attention. How does the greedy step affect training and decoding? It is also unclear how tricks in the paragraph above section 2.4 affect training and decoding. These questions should really be answered in [1]. Since the authors are extending their work and since these issues might cause training difficulties, it might be useful to look into these design choices.

The second question is about the window size $w$. Instead of imposing a fixed window size, which might not make sense for tasks with varying length segments such as the two in the paper, why not attend to the entire segment, i.e., from the current boundary to the previous boundary?

It is pretty clear that the model is discovering the boundaries in the utterance shown in Figure 2. (The spectrogram can be made more visible by removing the delta and delta-delta in the last subplot.) How does the MoCha attention look like for words whose orthography is very nonphonemic, for example, AAA and WWW?

For the experiments, it is intriguing to see that $w=2$ works best for speech recognition. If that's the case, would it be easier to double the hidden layer size and use the vanilla monotonic attention? The latter should be a special case of the former, and in general you can always increase the size of the hidden layer to incorporate the windowed information. Would the special cases lead to worse performance and if so why is there a difference?

[1] C Raffel, M Luong, P Liu, R Weiss, D Eck, Online and linear-time attention by enforcing monotonic alignments, 2017

---

> ### Author Response · Authors · 2017-12-27
> **Re: Great writing but lacks analysis**
>
> Thank you for your thorough review and thoughtful questions!  We're glad you found the paper easy to follow and the experiments convincing.  We've updated the paper to address your questions with additional experiments, and we provide some additional context below.
>
> > how far does using the expected context vector deviate from marginalizing the monotonic attention? The greedy step, described in the first paragraph of page 4, also has an effect on the produced attention. How does the greedy step affect training and decoding?
> Because we encourage the monotonic selection probabilities to be binary over the course of training by adding pre-sigmoid noise, these probabilities indeed tend to be 0 or 1 at convergence.  As a result, the greedy process is effectively equivalent to completely marginalizing out the alignment.  Note that we don't use the greedy step during training because we explicitly compute the probability distribution induced by the possible alignment paths.  We have added some wording to the paper to clarify these points.
>
> > It is also unclear how tricks in the paragraph above section 2.4 affect training and decoding. These questions should really be answered in [1]. Since the authors are extending their work and since these issues might cause training difficulties, it might be useful to look into these design choices.
> Indeed, [1] includes a "Practitioner's Guide" in Appendix G, which has discussion of how the sigmoid noise, weight norm, etc. can affect results.  We will add a reference to this practitioner's guide in the main text.  If you think it would be helpful, we can provide similar recommendations based on our own experiences in an appendix.
>
> > The second question is about the window size $w$. Instead of imposing a fixed window size, which might not make sense for tasks with varying length segments such as the two in the paper, why not attend to the entire segment, i.e., from the current boundary to the previous boundary?
> In fact, we also tried exactly what you suggested in early experiments!  We had planned to call this approach "MAtChA" for Monotonic Adaptive Chunkwise Attention.  As it turns out, it is possible to train this type of attention mechanism with an efficient dynamic program (analogous to the one used for monotonic attention and MoChA).  However, ultimately MAtChA did not outperform MoChA in any of our experiments.  In addition, the dynamic program used for training takes O(T^2U) memory instead of O(TU) memory (because you must marginalize out both the chunk start and end point, instead of just the end point), so we decided not to include it in the paper.  Prompted by your question, we've decided to put a discussion of MAtChA with a derivation of the dynamic program into the appendix.
>
> > The spectrogram can be made more visible by removing the delta and delta-delta in the last subplot.
> Great idea, we changed the figure to remove the delta and delta-delta features.
>
> > How does the MoCha attention look like for words whose orthography is very nonphonemic, for example, AAA and WWW?
> That's a very interesting point to discuss, so we added a note about this in the paper.  However, we were unable to find any such examples in the development set of the Wall Street Journal corpus, so we weren't able to study this directly.  Note that even for nonphonemic utterances, the attention alignment still tends to be monotonic - see for example Appendix A of "Listen, Attend and Spell" where a softmax attention model gives a monotonic alignment for your "AAA" example.
>
> > If that's the case, would it be easier to double the hidden layer size and use the vanilla monotonic attention?
> Thanks for this suggestion - indeed, using MoChA incurs a modest parameter increase (about 1% in our speech recognition experiments) because of the second independent energy function.  To address this difference, we ran an experiment where we doubled the attention energy function's hidden dimension in a monotonic attention model (similar in terms of parameter count to adding a second attention energy function) and halved this hidden dimension in a MoChA model.  In both cases, the change in performance was not significant over eight trials, implying that large gains achieved by MoChA were not caused by this change.  We added information about this experiment to the main text.

---

### Official Review · AnonReviewer2 · 2017-11-27
**Limited extension to previous work**

**Rating:** 6
**Confidence:** 4

**Review:**

The paper proposes an extension to a previous monotonic attention model (Raffel et al 2017) to attend to a fixed-sized window up to the alignment position. Both the soft attention approximation used for training the monotonic attention model, and the online decoding algorithm is extended to the chunkwise model. In terms of the model this is a relatively small extention of Raffel et al 2017.

Results show that for online speech recognition the model matches the performance of an offline soft attention baseline, doing significantly better than the monotonic attention model. Is the offline attention baseline unidirectional or bidirectional? In case it is unidirectional it cannot really be claimed that the proposed model's performance is competitive with an offline model.

My concern with the statement that all hyper-parameters are kept the same as the monotonic model is that the improvement might partly be due to the increase in total number of parameters in the model. Especially given that w=2 works best for speech recognition, it not clear that the model extension is actually helping. My other concern is that in speech recognition the time-scale of the encoding is somewhat arbitrary, so possibly a similar effect could be obtained by doubling the time frame through the convolutional layer. While the empirical result is strong it is not clear that the proposed model is the best way to obtain the improvement.

For document summarization the paper presents a strong result for an online model, but the fact that it is still less accurate than the soft attention baseline makes it hard to see the real significance of this. If the contribution is in terms of speed (as shown with the synthetic benchmark in appendix B) more emphesis should be placed on this in the paper.
Sentence summarization tasks do exhibit mostly monotonic alignment, and most previous models with monotonic structure were evaluated on that, so why not test that here?

I like the fact that the model is truely online, but that contribution was made by Raffel et al 2017, and this paper at best proposes a slightly better way to train and apply that model.

---
 The additional experiments in the new version gives stronger support in favour of the proposed model architecture (vs the effect of hyperparameter choices). While I'm still on the fence on whether this paper is strong enough to be accepted for ICLR, this version is certainly improves the quality of the paper.

---

> ### Author Response · Authors · 2017-12-27
> **Re: Limited extension to previous work**
>
> Thanks for your thorough review! We updated the paper to address your comments, and provide some additional discussion below.
>
> > Is the offline attention baseline unidirectional or bidirectional? In case it is unidirectional it cannot really be claimed that the proposed model's performance is competitive with an offline model.
> Thank you for pointing out this important distinction. The encoder in the softmax attention baseline is indeed unidirectional. We made this choice because using a unidirectional encoder is a prerequisite for an online model. We are interested in answering the question "how much performance is lost when using MoChA compared to using an offline attention mechanism?" so changing the encoder could conflate the difference in performance between the two models. The question "how much performance is lost when switching from a bidirectional encoder to a unidirectional encoder?" is interesting and important, but is orthogonal to what we are studying and has also been thoroughly considered in the past (e.g. in Graves et al. 2013). We have updated our wording to reflect exactly what we are studying and claiming.
>
> > My concern with the statement that all hyper-parameters are kept the same as the monotonic model is that the improvement might partly be due to the increase in total number of parameters in the model.
> This is also an important concern; however, the additional parameters required by MoChA compared to monotonic attention is tiny compared to the total number of parameters in the model because switching to MoChA amounts solely to adding a second attention energy function. For example, in our speech experiments, using MoChA increases the number of parameters by only 1.1%. To fully address this question, we ran experiments where we doubled the attention energy function's hidden dimension in a monotonic attention model and halved this hidden dimension in a MoChA model. This reconciles the difference in parameters in a natural way. In both cases, the change in performance was not significant over eight trials, implying that large gains achieved by MoChA were not caused by the change in parameter count. We added this information to the main text so that the comparison is clearer.
>
> > My other concern is that in speech recognition the time-scale of the encoding is somewhat arbitrary, so possibly a similar effect could be obtained by doubling the time frame through the convolutional layer.
> While we agree that increasing the receptive field of the convolutional layers could be helpful, we note that the recurrent layers in the encoder can in principle provide an arbitrarily long temporal context on their own. In addition, Bahdanau et al. 2014 implied that attention provides a more efficient way give the decoder greater long-term context. To test this empirically, we ran the suggested experiment where we doubled the convolutional filter size along the time axis in a monotonic attention-based model and found that it did not significantly change performance over eight trials. We added this experiment to the main text.
>
> > For document summarization the paper presents a strong result for an online model, but the fact that it is still less accurate than the soft attention baseline makes it hard to see the real significance of this.
> Our main rationale for including the document summarization experiment was to test MoChA in a setting where the input-output alignment was not monotonic. In terms of practicality, using MoChA would result in both a more efficient model (as you suggested) but could also allow for new applications such as online summarization. We added some additional clarification to the text as to our intentions behind this experiment.
>
> > Sentence summarization tasks do exhibit mostly monotonic alignment, and most previous models with monotonic structure were evaluated on that, so why not test that here?
> We avoided sentence summarization for the simple reason that it is an easy enough task that monotonic attention already matches the performance of softmax attention (see results in Raffel et al. 2017). We expect that MoChA would also match softmax attention's performance. Instead, we chose to try it on the more difficult (and more realistic) setting of CNN/daily mail. We included this discussion in the text of our paper to further motivate our experiment.
>
> > this paper at best proposes a slightly better way to train and apply that model.
> We consider MoChA to be a conceptually simple but remarkably effective improvement to monotonic attention. This is backed up by our experimental results, showing that we are able to significantly beat monotonic attention in settings where the alignment is monotonic (speech) and nonmonotonic (summarization). We see the simplicity of implementing MoChA on top of monotonic attention as a strength of our approach, in that it allows researchers and practitioners to easily leverage it.

---

> > ### Comment · AnonReviewer2 · 2018-01-01
> > **Thanks for your detailed response**
> >
> > Thanks for your detailed response and additional experiments and clarifications.

---

### Official Review · AnonReviewer3 · 2017-11-27
**Review of "Monotonic Chunkwise Attention"**

**Rating:** 8
**Confidence:** 4

**Review:**

This paper extends a previously proposed monotonic alignment based attention mechanism by considering local soft alignment across features in a chunk (certain window).

Pros.
- the paper is clearly written.
- the proposed method is applied to several sequence-to-sequence benchmarks, and the paper show the effectiveness of the proposed method (comparable to full attention and better than previous hard monotonic assignments).
Cons.
- in terms of the originality, the methodology of this method is rather incremental from the prior study (Raffel et al), but it shows significant gains from it.
- in terms of considering a monotonic alignment, Hori et al, "Advances in Joint CTC-Attention based End-to-End Speech Recognition with a Deep CNN Encoder and RNN-LM," in Interspeech'17, also tries to solve this issue by combining CTC and attention-based methods. The paper should also discuss this method in Section 4.

Comments:
- Eq. (16): $j$ in the denominator should be $t_j$.

---

> ### Author Response · Authors · 2017-12-27
> **Re: Review of "Monotonic Chunkwise Attention"**
>
> Thank you for your review! We are glad you found the paper clearly written, and that you were convinced by our experimental evaluation.  Addressing your specific comments:
>
> > in terms of the originality, this method is rather incremental from the prior study (Raffel et al)
> We would argue that the strength of our model demonstrates that this is not an incremental result; specifically, we saw a roughly 20% relative improvement compared to monotonic attention in terms of both the word error rate on speech recognition and ROUGE-2 on document summarization.  Further, on speech recognition, we showed for the first time that an online attention mechanism could match the performance of an (offline) softmax attention mechanism, which opens up the possibilities of using this framework in online settings.  While MoChA can be seen as a conceptually straightforward extension of Monotonic Attention, we actually see that as a benefit of the approach - it would potentially be less impactful if achieving these benefits required a complicated modification to the seq2seq framework.  We have added some language to emphasize this at the end of Section 1.
>
> > - in terms of considering a monotonic alignment, Hori et al, "Advances in Joint CTC-Attention based End-to-End Speech Recognition with a Deep CNN Encoder and RNN-LM," in Interspeech'17, also tries to solve this issue by combining CTC and attention-based methods. The paper should also discuss this method in Section 4.
> Thank you for bringing this paper to our attention.  The primary difference between that paper and ours it that it still uses an offline softmax attention mechanism, so could not be used in online settings.  However, it provides promising evidence that our approach could be combined with CTC to achieve further gains in online settings.  We've added this reference and some discussion of it to our related work section.
>
> > Eq. (16): $j$ in the denominator should be $t_j$.
> Excellent catch, thank you! We have fixed this.

---

### Decision · Program_Chairs · 2018-01-29
**ICLR 2018 Conference Acceptance Decision**

**Decision:**

Accept (Poster)

**Comment:**

This clearly written paper describes a simple extension to hard monotonic attention -- the addition of a soft attention mechanism that operates over a fixed length window of inputs that ends at the point selected by the hard attention mechanism.  Experiments on speech recognition (WSJ) and on a document summarization task demonstrate that the new attention mechanism improves significantly over the hard monotonic mechanism.  About the only "con" the reviewers noted is that the paper is a minor extension over Raffel et al., 2017, but the authors successfully argue that the strong empirical results render this simplicity a "pro."